# Magnetoelectric MEMS Magnetic Field Sensor Based on a Laminated Heterostructure of Bidomain Lithium Niobate and Metglas

**DOI:** 10.3390/ma16020484

**Published:** 2023-01-04

**Authors:** Andrei V. Turutin, Elena A. Skryleva, Ilya V. Kubasov, Filipp O. Milovich, Alexander A. Temirov, Kirill V. Raketov, Aleksandr M. Kislyuk, Roman N. Zhukov, Boris R. Senatulin, Victor V. Kuts, Mikhail D. Malinkovich, Yuriy N. Parkhomenko, Nikolai A. Sobolev

**Affiliations:** 1Laboratory of Physics of Oxide Ferroelectrics, National University of Science and Technology MISiS, 119049 Moscow, Russia; 2Department of Physics and I3N, University of Aveiro, 3810-193 Aveiro, Portugal; 3Mapper LLC, Volgogradsky Pr. 42 k. 5, 109316 Moscow, Russia; 4JSC ‘‘Giredmet’’, 2 Elektrodnaya Str., 111524 Moscow, Russia

**Keywords:** MEMS, magnetic sensors, metglas, bidomain lithium niobate, magnetoelectric effect, thin films, microblasting

## Abstract

Non-contact mapping of magnetic fields produced by the human heart muscle requires the application of arrays of miniature and highly sensitive magnetic field sensors. In this article, we describe a MEMS technology of laminated magnetoelectric heterostructures comprising a thin piezoelectric lithium niobate single crystal and a film of magnetostrictive metglas. In the former, a ferroelectric bidomain structure is created using a technique developed by the authors. A cantilever is formed by microblasting inside the lithium niobate crystal. Metglas layers are deposited by magnetron sputtering. The quality of the metglas layers was assessed by XPS depth profiling and TEM. Detailed measurements of the magnetoelectric effect in the quasistatic and dynamic modes were performed. The magnetoelectric coefficient |α_32_| reaches a value of 492 V/(cm·Oe) at bending resonance. The quality factor of the structure was Q = 520. The average phase amounted to 93.4° ± 2.7° for the magnetic field amplitude ranging from 12 to 100 pT. An AC magnetic field detection limit of 12 pT at a resonance frequency of 3065 Hz was achieved which exceeds by a factor of 5 the best value for magnetoelectric MEMS lead-free composites reported in the literature. The noise level of the magnetoelectric signal was 0.47 µV/Hz^1/2^. Ways to improve the sensitivity of the developed sensors to the magnetic field for biomedical applications are indicated.

## 1. Introduction

The magnetoelectric (ME) effect occurs in a material when electric polarization is produced by an externally applied magnetic field (direct ME effect), and vice versa, a change in the magnetization occurs in an electric field (converse ME effect) [1]. Such properties have a high practical potential for application in science and technology. Based on ME materials, it is possible to manufacture several devices with unique properties, such as, for example, random-access ME memory, ME sensors of magnetic fields and electrical currents, etc. [2]. Therefore, the search for new composite ME materials, the study of the ME effect in them, as well as the miniaturization of the respective devices have priority and are urgent tasks in the creation of novel electronics.

A key trend in modern medicine and physiology is the transition to non-invasive methods of diagnosing diseases. In particular, the study of magnetic fields in the central nervous system is called magnetoencephalography, and in the region of the heart muscle–magnetocardiography. The need to detect ultraweak magnetic fields requires the use of superconducting quantum interferometers (SQUID), which greatly complicates the design of the device due to the need for a cryogenic system and, ultimately, leads to a significant increase in its cost, preventing wide dissemination of the technique in ordinary clinical institutions. Thus, recently there has been a large demand for cheap, compact and sensitive magnetic sensors for biomedical applications [3] which require the ability to detect very weak fields with amplitudes lower than 10 pT at frequencies below 100 Hz [4]. For this purpose, a new type of ME sensor based on composite materials has been intensely investigated, potentially offering a sensitive, low-cost and room-temperature operation [2]. In particular, simple ME laminate heterostructures containing mechanically coupled magnetostrictive (MS) and piezoelectric (PE) layers are capable of producing large voltages in response to weak magnetic fields [5]. In this regard, ME sensors (as well as any other devices for detecting weak signals, including SQUIDs) require special measures to reduce intrinsic and external noise. To date, the magnetic field detectivity of the best sensors based on bulk composite ME materials ranges from 100 fT/Hz^1/2^ to a few pT/Hz^1/2^ depending on frequency [6,7]. This sensitivity is sufficient for the reliable detection of magnetic fields induced by the α-rhythm currents in the brain. To study the activity of the cerebral cortex, it is necessary to measure magnetic fields 1 to 2 orders of magnitude weaker. To date, a sensor based on composite ME materials with this sensitivity has not yet been implemented. Furthermore, such structures have tens of mm in length, and it is difficult to employ them for mapping magnetic fields with a high spatial resolution.

An important direction from the point of view of the practical use of ME composites is the miniaturization of structures with a gradual transition to the technology of microelectromechanical systems (MEMS) [8]. Most ME MEMS use the conventional technology steps of lithography and deposition of thin films of PE and MS materials onto a silicon substrate, which is then etched to form thin cantilevers. The main PE materials used for ME MEMS are AlN [9], PZT [10], AlScN [11], x-cut quartz [12].

Among magnetic materials for the MS phase, different types of amorphous metglas alloy are generally considered [13]. FeGaC [14] and FeCoC [15] thin films exhibit the highest piezomagnetic coefficients (q) of 10 ppm/Oe. However, for ME MEMS, FeCoSiB with a comparable q = 5 ppm/Oe is mostly used [13,16]. Magnetically soft metglas thin films are most interesting for detecting ultraweak magnetic fields due to the high values of the piezomagnetic coefficient in low magnetic fields, which makes it possible to create self-biased devices [17]. However, in the technology of deposition of MS metglas layers, there are several problems associated with the crystallization process in the films due to the heating of substrate during the magnetron sputtering, stresses in the sputtered film and associated magnetic anisotropies [16,18]. In FeCoSiB, MS properties strongly depend on the degree of amorphization of thin films [19,20].

The highest sensitivity of 300 pT to a DC magnetic field for ME MEMS was achieved using the delta-E effect for the AlN/(FeGaB/Al_2_O_3_)-based structure [21]. An increase in sensitivity to AC magnetic fields can be accomplished using the electromechanical resonance of the ME structure. By employing Al_0.73_Sc_0.27_N and (Fe_90_Co_10_)_78_Si_12_B_10_ thin films for PE and MS phase, respectively, a ME MEMS sensor has shown a limit of detection (LOD) of 60 pT/Hz^1/2^ at the resonance frequency f_r_ = 8185 Hz [11]. It has been shown that the noise increasingly dominating in the ME MEMS with decreasing size is of thermomechanical nature [22]. The first ME CMOS-MEMS integrated sensor array showed a LOD of 16 nT/Hz^1/2^ with low power consumption [23]. In the frequency range desirable for biomedical applications, ME MEMS sensors can measure magnetic fields with a sensitivity of 100 pT/Hz^1/2^ [13]. Urgent tasks are to discover new promising PE and MS materials and develop fabrication processes on the MEMS scale and more accurate detection systems with fast digital signal processing.

Asymmetrical bi-layered systems with bimorph PE components comprising two oppositely poled layers along the thickness direction are known to exhibit particularly large ME coefficients at low-frequency bending resonance [24]. The simplest example of such a bi-layered system is a laminated bimorph. Commonly, PE bimorphs have been mostly fabricated by bonding or sintering together lead-containing PE plates with opposed polarizations [24,25]. Those bimorphs, however, exhibit large mechanical losses and creep effects. Nakamura et al. [26] and Kubasov et al. [27] developed new promising techniques to directly engineer two polarization domains in a single bulk ferroelectric crystal (bidomain), thus excluding any bonded interfaces [28,29]. The absence of an intermediate viscous glue layer between macrodomains results in high thermal stability and linearity of the deformation-to-voltage conversion, which makes them ideal for precise sensing, actuation and energy harvesting [30,31,32,33,34]. We proposed the use of bidomain lithium niobate (b-LN) single crystals [7,35,36,37] to decrease the operation frequency and partially suppress extrinsic and intrinsic noises in ME composites. Our best samples exhibited a record detectivity among ME composites as low as 92 fT/Hz^1/2^ at a bending resonance frequency of 6862 Hz [7]. Furthermore, bulk crystals of b-LN with a y + 128°-cut possess a relatively high value of |d/ε| = 0.55 pm/V [38]. ME heterostructures based on b-LN y + 140°-cut/metglas demonstrate a high Q-factor of 1100 [7]. Spetzler et al. recently showed that the LOD for ME magnetic field sensors utilizing the delta-E effect is proportional to Q^−3/2^ independently of the signal frequency if the thermomechanical noise dominates [9].

In this work, we present a composite ME MEMS cantilever based on a b-LN y + 128°-cut/thin-film-Fe_70_Co_8_Si_12_B_10_ heterostructure. A thin LN cantilever was manufactured by the microblasting technique as a part of the original LN substrate. Thus, a near-ideal cantilever structure was realized, which allows us to keep all the advantages of bulk b-LN crystals such as the high Q-factor, low tangential losses, high |d/ε| ratio and low-frequency bending mode. Furthermore, there is no mismatch between the thermal expansion coefficients of the substrate and the PE layer.

## 2. Materials and Methods

### 2.1. LN Thinning Method

Single-domain LN wafers with a y + 128°-cut, purchased from The Roditi International Corporation Ltd. (London, UK), were used as the PE component of the ME heterostructure under study. A negative photoresist film supplied by the HARKE Group (Mülheim an der Ruhr, Germany) was applied to the LN plate and exposed to UV light to obtain a rectangular window for the subsequent material removal. After the development of the protective photoresist mask, the initial wafer having a thickness of 500 µm was thinned down to ca. 80 µm inside a rectangular area by the microblasting technique using an automated system by Comco Inc. (Burbank, CA, USA). Then, the last two procedures were repeated in a Π-shaped area to obtain a MEMS cantilever structure. Aluminium oxide abrasive PD1009 supplied by Comco was used for microblasting. To determine the optimal regime, LN layers were removed in multiple subsequent steps at an air pressure in the abrasive feeding system varying from 450 kPa to 600 kPa and an increment of 50 kPa. The roughness and etching depth of LN wafers were measured on a Bruker Contour GT-K optical profilometer using vertical scanning interferometry.

### 2.2. Sputtering of Metglas Films

The electrodes and the MS layer were sputtered using the magnetron system SUNPLA-40TM (Seoul, Republic of Korea) from multi-element targets of nichrome (Cr_20_Ni_80_) and metglas (Fe_70_Co_8_Si_12_B_10_) having diameters of 50 mm. The synthesis processes were performed in an Ar atmosphere at a chamber pressure of 0.5 Pa. A metal mask was attached to the substrate during the processes to avoid electrical short-circuiting between the top and bottom electrodes. First, 100 nm thick nichrome layers serving as electrodes were deposited at both sides of the MEMS cantilever using DC magnetron sputtering. Then, the MS metglas layer was deposited on one side of the cantilever by RF magnetron sputtering. The power of the magnetron and the distance between the metglas target and the cantilever (200 W and 45 mm, respectively) were selected to provide the highest possible growth rate of the film with minimal heating of the substrate to avoid crystallization of the MS layer. To improve the technology of metglas magnetron sputtering, a three-steps procedure was implemented. The deposition rate was chosen as the main factor influencing the quality of the obtained films. A gradual change in deposition rate by adjusting the ongoing synthesis process made it possible to obtain better characteristics of magnetic films at higher deposition rates. ME MEMS structures were obtained with the total duration of the metglas sputtering process of 600 min, while the film with a thickness of 2 µm was deposited.

### 2.3. Investigation of Metglas Films by TEM and XPS

X-ray photoelectron spectroscopy (XPS) analysis was carried out with a VersaProbe II spectrometer (ULVAC-PHI) equipped with a monochromatic AlKα radiation source (hν = 1486.6 eV). A focused X-ray beam with a diameter of 200 µm operated at 50 W was used to excite photoemission. Survey spectra obtained in the range of 0–1400 eV with a low energy resolution were used to determine the total elemental composition. The high-resolution (HR) narrow scans were performed to determine the chemical state of Fe, Si and B. HR spectra were approximated by a nonlinear least-squares method using mixed Gaussian–Lorentzian line shapes for B 1s and Si 2s bands, asymmetric line shape for the metallic Fe 2p3 peak and multiple peaks for the oxidized Fe 2p3. The binding energy (BE) scale was calibrated using Au and Cu metals (Au 4f–83.96 eV and Cu 2p3/2–93.62 eV). When surface charging took place, C 1s (284.8 eV) was used as a reference. Depth profiles were obtained by alternating 2 keV Ar+ ion sputtering and recording XPS spectra (O 1s, Fe 2p1, Co 2p1, Si 2p and B 1s spectra). Atomic fractions were calculated by the respective peak areas using Physical Electronics (PHI) sensitivity factors corrected by the transmission function of the analyzer.

Structural studies were performed at a transmission electron microscope (TEM) JEM 2100 (JEOL, Akishima City, Tokyo, Japan) with an accelerating voltage of 200 kV. For the TEM study, the samples were thinned using a focused ion beam Strata FIB 201 System (FEI Company, Blackwood NJ, USA)

### 2.4. Bidomain LN Preparation

After thinning the LN plates, a ferroelectric bidomain was formed in the crystals. The structures were cleaned after the microblasting process and annealed for 5 h at a temperature of 1100 °C in an air atmosphere, thus, conditions were created for the formation of a bidomain ferroelectric structure of the head-to-head type. The method we used is based on the evaporation of lithium oxide out of the surface of the LN crystal [27] (so-called out-diffusion annealing). The crystals were placed in a muffle furnace on a sapphire wafer using narrow and thin (0.5 mm) sapphire spacers, thus a symmetrical out-diffusion of Li_2_O occurred. After annealing, an angle lap of the reference console was prepared and etched in an HF:HNO_3_ = 2:1 (vol.) mixture for the visualization of the domain structure according to Ref. [39] as is seen in Figure 1.

### 2.5. ME Effect Measurements

Measurements of the quasi-static and dynamic ME coefficients were carried out to study the ME effect in the ME MEMS structures. The external magnetic field was created using self-made Helmholtz coils, consisting of two pairs of coaxial coils and capable of simultaneously applying AC and DC magnetic fields. A lock-in amplifier (Zurich instruments, MFLI, Zurich, Switzerland) was used to read the signal from the sample and as a precision signal source for the Helmholtz coils. Quasi-static measurements were carried out in the amplitude range from 0 Oe to 10 Oe with a step of 0.5 Oe when applying an AC magnetic field with an amplitude of 0.1 Oe at a frequency of 117 Hz. In the dynamic mode, the ME coefficient was measured as a function of the frequency of the AC magnetic field with an amplitude of 0.1 Oe whereas an optimal DC magnetic field was applied to the ME sample. To measure the sensitivity of the ME MEMS structure, the ME voltage V_ME_ was detected as a function of the amplitude of the AC magnetic field which varied from 1 pT to 10 nT at the frequency of the electromechanical resonance.

## 3. Results and Discussions

### 3.1. LN Thinning Results

During the determination of the optimal air pressure for LN thinning, the average decrement of the thickness of the cantilever at each step (machining rate) as well as the average (R_a_) and total (R_t_) roughness of its surface were measured using an optical interferometer. It was found that the increase of the air pressure in the abrasive feeding system in the range between 450 kPa and 550 kPa enhances the machining rate only slightly with a simultaneous decrease of the total roughness R_t_ (Table 1). Further growth of the air pressure leads to a drastic increase in the machining rate and both the roughness values. Such an effect is not desirable as an enhanced roughness is evidence of the surface damage which can destroy the cantilever. Therefore, the pressure of 550 kPa was chosen for further microblasting processes.

### 3.2. Composition and Structure of Metglas Films

Metglas films deposited by magnetron sputtering from a Fe_70_Co_8_Si_12_B_10_ target should be amorphous and free of metal, silicon, and boron oxides. The XPS provides control over the oxygen and oxides content, and TEM evaluates the degree of crystallinity of the metglas layers.

The technology of metglas layer deposition was adjusted according to XPS and TEM resutls. The process of optimizing deposition technology can be divided into three stages. Test samples produced under suboptimal conventional process conditions (first stage) at low deposition rates (<0.3 nm/min) and unstable vacuum environment exhibited very high oxygen concentrations on the order of 50–60 at. %. Figure 2 shows HR spectra in the Fe 2p, Si 2s and B 1s spectral ranges. The curve fitting of the Fe 2p doublet demonstrates a complex overlap of Fe_3_O_4_ ground-state multiplet peaks [40] with a small asymmetric metallic iron peak, i.e., most of the iron is oxidized. The Si 2s peak, located at 153.4 eV, corresponds to oxidized silicon [41]. The B 1s peak is chemically shifted to 192.4 eV, which means the existence of the B_2_O_3_ oxide [42].

Figure 3 shows a bright-field TEM image of the lamella edge of a metglas-on-silicon sample produced by FIB at the first stage of the technology improvement. Crystalline particles are visible in the image (Figure 3b), and the selective area electron diffraction (SAED) analysis shows a significant match with the α-Fe phase (Figure 3c). In addition to crystalline particles of α-Fe, the film contains an amorphous phase. This conclusion is based on the presence of an amorphous halo in the electron diffraction pattern together with reflections from α-Fe.

Based on the data from XPS elemental analysis and TEM structural analysis, we can draw the following conclusions. The proportion of the amorphous phase prevails in quantity over the crystalline phase. The largest part of iron by volume is in an amorphous and oxidized state. The formation of the crystalline phase is most likely associated with the technological conditions of the film deposition. A significant amount of oxygen, 50–60 at. %, and the formation of the α-Fe crystalline phase indicate unsuitable technological conditions for the formation of a homogeneous amorphous structure.

At the second stage of process optimization, thorough cleaning of all internal parts of the magnetron sputtering chamber and check-up of all possible leakages for external air were undertaken. Despite these measures, the oxygen concentration was reduced only to 30–40 at. %, which indicates a strong tendency of the growing metglas film to oxidation. Figure 4a shows the depth profile of iron in a sample obtained under improved vacuum conditions and after some design upgrades.

The deposition time was 120 min, which yields a thickness of the metglas films of 40 to 60 nm and a corresponding deposition rate of 0.3 to 0.5 nm/min. The HR spectra in Figure 5 show only partial oxidation of iron, complete oxidation of boron and the appearance of a weak peak (150.2 eV) from elemental silicon against a strong peak from oxidized silicon at 153.4 eV.

At the third stage of the process development, a significant suppression of the oxygen incorporation was achieved. We concluded that increasing the sputtering rate of the Fe_70_Co_8_Si_12_B_10_ target reduced the effect of a constant residual amount of oxygen in the chamber on the composition of the synthesized layer. Films with a thickness of about 1000 nm with an oxygen content of 5 at. % were obtained for a deposition time of 360 min, i.e., the deposition rate was about 2.8 nm/min. Figure 4b shows the composition distribution up to a depth of 140 nm from the surface for a coating about 1000 nm thick. The oxygen fraction amounts between 4 and 6 at. % throughout the entire depth of Ar+ etching.

Here it should be noted that, despite the difficulties of quantitative XPS analysis associated with the superposition of the Fe and Co lines and the preferential sputtering effect, an adequate comparative estimate of the oxygen content was obtained. An analysis of the HR spectra confirms this result. The Fe 2p (706.7 eV) and B 1s (187.8 eV) spectra of the ~1000 nm thick sample with 5 at. % oxygen content show metallic iron and elemental boron with no detectable presence of oxides, Figure 6. Only the Si 2s spectrum (150.6 eV) demonstrates elemental silicon with a minor (≤10%) contribution from oxidized silicon.

Figure 7 shows the homogeneous amorphous structure of the metglas produced at the third stage of the technology improvement, which is confirmed by the electron diffraction pattern shown in Figure 7c, and the bright-field image in Figure 7b. Changing the process conditions and a significant increase in the metglas film deposition rate from <1 nm/min to 8 nm/min led to the formation of an amorphous material with a uniform distribution of elements (Figure 4). Compared with the first and second stages of the technology improvement, reducing the amount of oxygen to ≈ 5 at. % at the third stage led to the fact that no oxidized states of the main elements in the metglas film were detected anymore by XPS.

### 3.3. ME Measurements and Sensitivity to AC Magnetic Field

The ME measurements were carried out at the laminated ME MEMS composite “b-LN/Fe_70_Co_8_Si_12_B_10_”. A photo and schematic layer sequence of the ME MEMS structure are shown in Figure 8a. The b-LN cantilever was 80 µm thick, 4.5 mm long, and 3 mm wide. The thickness of the Fe_70_Co_8_Si_12_B_10_ MS layer was 2 μm. Metglas was deposited on the polished side of the b-LN substrate. A 100 nm thick nichrome (Cr_20_Ni_80_) contact was deposited onto the opposite side of the cantilever. Measurement results of the quasi-static and dynamic ME coefficients are shown in Figure 8b.

The maximum quasi-static |α_32_| = 1.2 V/(cm·Oe) is achieved at an optimal DC magnetic field of 6 Oe. The low value of the latter for the deposited Fe_70_Co_8_Si_12_B_10_ metglas indicates the amorphous state of the material and the absence of the crystalline α-Fe phase. In previously published papers, where a similar metglas composition was used as an MS layer in ME structures, the optimal magnetic field varied between 9 Oe and 16 Oe [11,17,43]. However, the optimal field value also depends on the demagnetizing factor of the structure. Therefore, for a correct comparison, it is necessary to take into account the linear dimensions of the MS layer. With an increase in the length of the MS layer, the optimal magnetic field will shift to lower values [37,44].

In dynamic ME effect measurements, |α_32_| reaches the maximum value of 492 V/(cm·Oe) at the bending resonance frequency. The quality factor of the structure was Q = 520. The sensitivity to AC magnetic field was measured at resonance and the results are presented in Figure 9.

The sensitivity of the ME MEMS structure to the magnetic field was measured as the V_ME_ dependence on the amplitude of the AC magnetic field. The obtained results were expressed in terms of the voltage noise spectral density which corresponds to the root mean square of the measured voltage amplitude normalized by the square root of the bandwidth of the input filter in units of V/Hz^1/2^. The bandwidth of the input low-pass filter was equal to 1 Hz. The linear approximation used the least squares method and considered error bars. For each experimental point in Figure 9, the standard deviation was calculated from 1000 measured values at an applied magnetic field.

The detection limit to AC magnetic field was 12 pT at a frequency of 3065 Hz. The noise level of the ME signal was 0.47 µV/Hz^1/2^. The V_ME_ value in a wide range of magnetic field amplitudes (from 12 pT to 100 nT) demonstrates good linearity with R^2^ = 0.998. Also, Figure 9 shows the phase of the V_ME_ signal. The average phase value was 93.4° ± 2.7° for the magnetic field amplitude ranging from 12 pT to 100 pT. The V_ME_ signal phase has a large spread at magnetic field amplitudes below 12 pT due to the achievement of the magnetic field detection limit by the ME MEMS structure.

Table 2 shows the values of the resonant ME coefficient, conversion coefficient (equal to α *t*, where *t* is the thickness of the PE phase), quality factor and sensitivity to AC magnetic field at the respective resonance frequency for various lead-free ME MEMS composites in comparison with our results.

Our ME MEMS “b-LN/Fe_70_Co_8_Si_12_B_10_” laminate has a ME coefficient comparable with that of the AlN-based structures and exhibits the highest conversion coefficient. The latter is used to calculate the sensitivity limit of an ME material to a magnetic field as a ratio of the voltage noise level of the ME sample and the converse coefficient. Importantly, the sensitivity to the magnetic field of the ME MEMS “b-LN/Fe_70_Co_8_Si_12_B_10_“ at a relatively low resonance frequency exceeds by a factor of 5 the best value obtained for the “poly-Si/AlScN/FeCoSiB” ME MEMS. The main limitation to the magnetic field sensitivity is the environmental vibration signals readily exciting the bending resonance together with the AC magnetic field. In further devices the influence of the vibrational noise will be significantly decreased by using tuning-fork approach [46].

## 4. Conclusions

In summary, we report a novel technology for the creation of a lead-free ME MEMS based on the b-LN/Fe_70_Co_8_Si_12_B_10_ heterostructure comprising bidomain lithium niobate (b-LN) and metglas layers. The b-LN cantilever was manufactured using the microblasting technique. The optimal pressure of 550 kPa for the supply of the aluminium oxide abrasive increased the machining rate and saved low roughness values of R_a_ = 0.8 µm and R_t_ = 10.05 µm. The thinned LN crystal was obtained as a part of the original LN substrate; thus, a near-ideal cantilever structure was implemented. We formed a ferroelectric bidomain structure in 80 µm thick y + 128°-cut LN crystals. An amorphous metglas thin film with a uniform distribution of elements was sputtered using RF magnetron at a 3.2 nm/min deposition rate. The ME MEMS composite exhibited a reasonably high ME coefficient and Q-factor. The AC magnetic field detection limit at a resonance frequency of 3065 Hz was 12 pT which exceeds by a factor of 5 the best value obtained for the ME MEMS based on the poly-Si/AlScN/FeCoSiB composite. In the future, to further improve the sensitivity, we intend to use an ME composite structure in the form of a tuning fork previously developed by us: a multiple increase in the sensitivity to a magnetic field was achieved in this way thanks to the cancellation of the vibrational noise [46,47]. Nevertheless, signals gathered by ME MEMS sensors are necessarily noisy, thus additional measures for data collection and processing such as active shielding, digital filtering, noise cancellation, adaptive averaging, etc. must be applied [6]. Moreover, the implementation of modulation techniques based on frequency conversion allows for minimizing the residual effect of acoustic noise on the sensor [48]. This promising technique together with data acquisition based on a high-resolution ADC and sophisticated signal processing were recently applied to the robust detection of magnetic signals from a human heart [6]. Thus, further efforts will be focused on increasing the thickness ratio of the MS to PE phases to enhance the ME effect using tuning-fork-like heterostructures and improving the detection system to measure ultra-weak biomagnetic signals.

## Figures and Tables

**Figure 1 materials-16-00484-f001:**
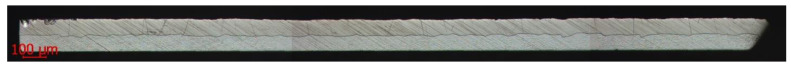
Micro photo of the etched angle lap of a y + 128°-cut LN crystal with bidomain structure.

**Figure 2 materials-16-00484-f002:**
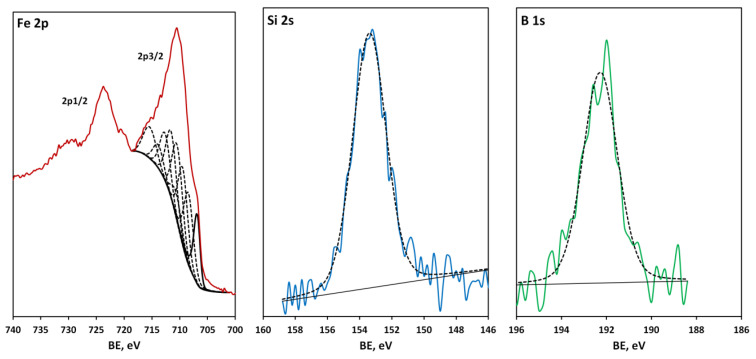
(Color online) Fe 2p, Si 2s and B 1s spectra of a metglas-on-silicon sample produced at the first stage of the technology improvement. Experimental spectra (solid colour lines) were fitted using the bands which are depicted by black dashed lines, save the band at 706.7 eV assigned to metallic iron, highlighted with a solid black line.

**Figure 3 materials-16-00484-f003:**
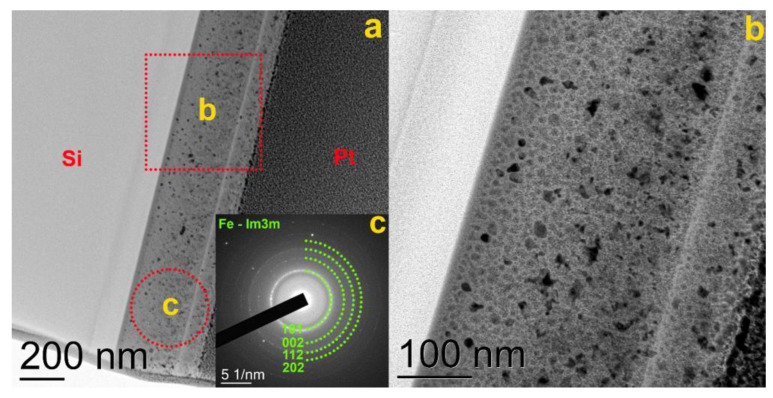
(**a**) TEM image of a metglas-on-silicon film produced at the first stage of the technology improvement; (**b**) enlarged lamella area; (**c**) SAED image with the specified SAED acquisition area.

**Figure 4 materials-16-00484-f004:**
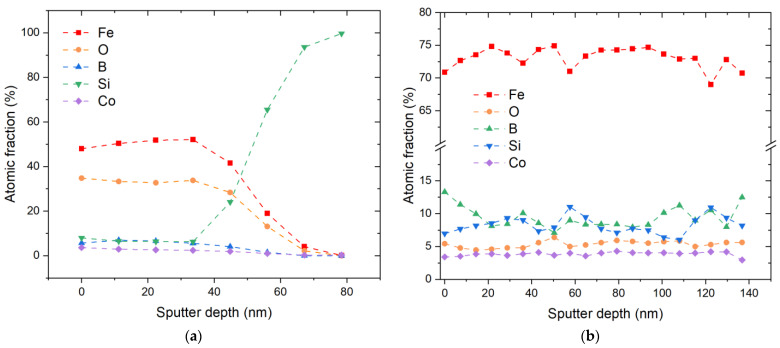
(Color online) Depth profile (**a**) of a 50 nm thick metglas layer on silicon, produced at the second stage of the technology improvement, and (**b**) of a 1000 nm thick sample, produced at the third stage of the technology improvement.

**Figure 5 materials-16-00484-f005:**
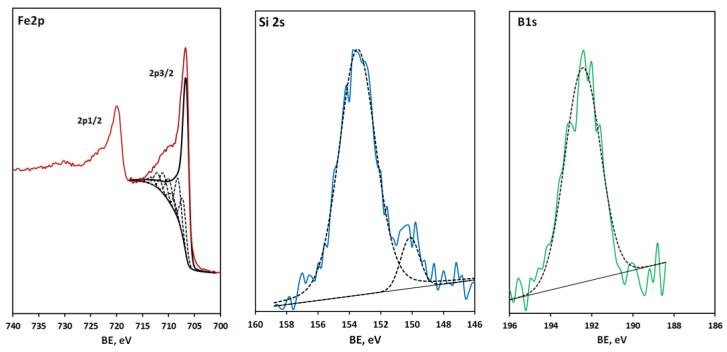
Fe 2p, Si 2s and B 1s spectra of a metglas-on-silicon sample, produced at the second stage of the technology improvement. Experimental spectra (solid colour lines) were fitted using the bands which are depicted by black dashed lines, save the band at 706.7 eV assigned to metallic iron, highlighted with a solid black line.

**Figure 6 materials-16-00484-f006:**
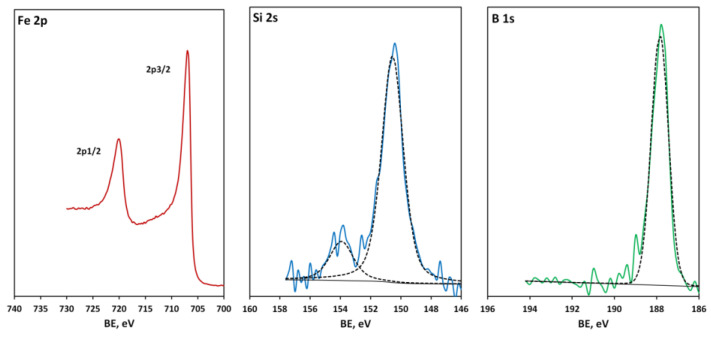
Fe 2p, Si 2s and B 1s spectra of a metglas-on-silicon sample produced at the third stage of the technology improvement. The experimental Si 2s and B 1s spectra (solid colour lines) were fitted using individual bands (black dashed lines). The experimental Fe 2p spectrum (red line) corresponds to metallic iron without an oxidized component.

**Figure 7 materials-16-00484-f007:**
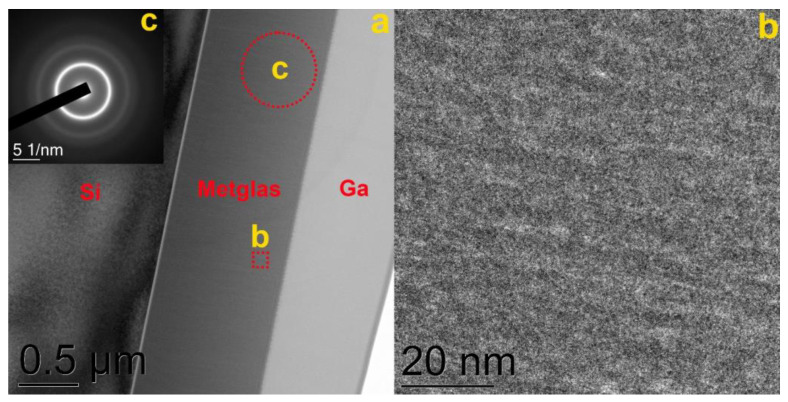
(**a**) TEM images of a metglas film produced at the third stage of the technology improvement comprising 5% oxygen; (**b**) enlarged lamella area; (**c**) SAED image with the specified SAED acquisition area.

**Figure 8 materials-16-00484-f008:**
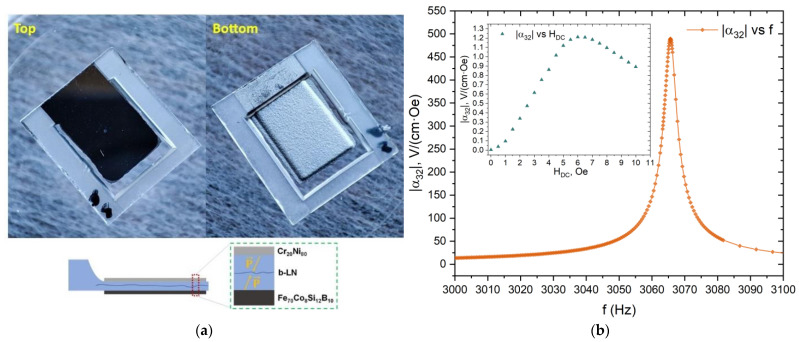
(**a**) Photo of the top and bottom view of the ME MEMS sample and schematic layer structure, where the arrows indicate spontaneous polarization vectors in the head-to-head bidomain LN crystal. (**b**) Dynamic direct ME coefficient |α_32_| measured as a function of the frequency of AC magnetic field with an amplitude of 0.1 Oe and the optimal bias field H = 6 Oe. The inset shows the quasi-static in-plane direct ME coefficient |α_32_| measured as a function of the DC magnetic field.

**Figure 9 materials-16-00484-f009:**
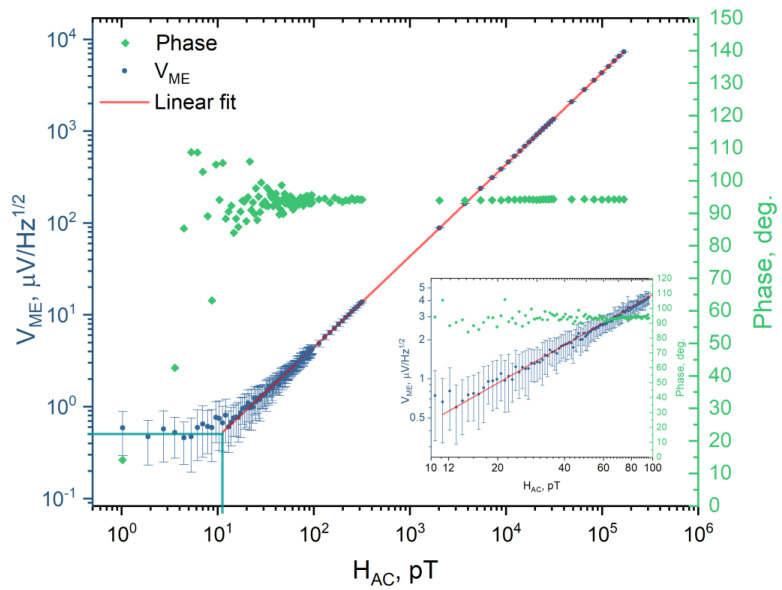
ME voltage (V_ME_) and signal phase measured as a function of the AC magnetic field amplitude at the bending resonance frequency (3065 Hz). Error bars indicate the standard deviation for the measured V_ME_ values. The horizontal and vertical straight blue lines show the voltage noise level and magnetic field detection limit, respectively. The inset represents an enlarged portion of the graph in the range from 10 pT to 100 pT.

**Table 1 materials-16-00484-t001:** LN thinning results.

Air Pressure in the Abrasive Feeding System, kPa	R_a_, µm	R_t_, µm	Machining Rate/Scan, µm
450	0.75	13.10	18.43
500	0.82	12.14	20.23
550	0.80	10.05	22.69
600	2.33	31.96	27.21

**Table 2 materials-16-00484-t002:** Comparison of lead-free ME MEMS structures.

ME MEMS Structure	α, V/(cm·Oe)	Conversion Coefficient, V/Oe	f_r_, Hz	Q-Factor	Sensitivity, pT/Hz^1/2^	Ref.
SiO_2_/AlN/FeCoSiB	800	0.08	2400	74	100	[43]
poly-Si/AlN/FeCoSiB	734	0.0734	8965	592	62	[11]
poly-Si/AlScN/FeCoSiB	1334	0.1334	8002	600	60	[11]
poly-Si/AlN/FeCoSiB	756	0.0378	7760	2217	175	[45]
b-LN/Fe_70_Co_8_Si_12_B_10_	492	3.936	3065	520	12	This work

## Data Availability

Not applicable.

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
