# Peer review of "Magnetoelectric MEMS Magnetic Field Sensor Based on a Laminated Heterostructure of Bidomain Lithium Niobate and Metglas"

_materials, 2023, doi:10.3390/ma16020484_

Round 1

Reviewer 1 Report

Turutin et al. have presented the manuscript titled: Magnetoelectric MEMS magnetic field sensor based on a laminated heterostructure of bidomain lithium niobate and metglas. Overall presentation of the article is good, but there require few modifications before being publish, suggestions are as follow;

1.      Abstract section is not strong as compared to the study; authors should highlight their results values in the abstract to attract the readers, like Depth profile, ME coefficient, ME voltage (VME) and signal phase measured as a function of the AC magnetic field amplitude, etc.

2.      Authors are suggested to add the elemental mapping or EDX analysis of along with Figure 3.

3.      Line 232-233, “The Si 2s peak, located at 153.4 eV, corresponds to SiO2. The B 1s peak is chemically shifted to 192.4 eV, which means the existence of the B2O3 oxide.” Please mention the references for these measurements.

4.      In the discussion of the Figure 2, 3, and 4 authors have just illustrated the results, without describing the reasoning.

5.      Overall the research work is very well explained, please make the conclusion precise and remove the extra discussion sentences from conclusion.

Author Response

Reviewer: Turutin et al. have presented the manuscript titled: Magnetoelectric MEMS magnetic field sensor based on a laminated heterostructure of bidomain lithium niobate and metglas. Overall presentation of the article is good, but there require few modifications before being publish, suggestions are as follow;

(1) Abstract section is not strong as compared to the study; authors should highlight their results values in the abstract to attract the readers, like Depth profile, ME coefficient, ME voltage (VME) and signal phase measured as a function of the AC magnetic field amplitude, etc.

Response:

Thank you for your elaborate comment. We added the necessary information to the abstract and highlighted it.

(2) Authors are suggested to add the elemental mapping or EDX analysis of along with Figure 3.

Response:

Unfortunately, our TEM is not equipped with a scanning transmission electron system (STEM), which is necessary for mapping or accurate analysis along the line. We can only use EDX by points. We did this for the sample shown in fig. 3 and the results are presented in the table below.

Table. EDX analysis by points.

L (nm)

O (at%)

Si (at%)

Fe (at%)

Co (at%)

0

30.5

6.3

54.7

8.3

90

25.2

6.9

58.4

9.4

180

24.6

6.8

59.0

9.5

270

24.2

6.8

59.8

9.0

360

22.9

7.0

60.2

9.7

The EDX method does not detect well light elements and has a large error in the quantitative analysis of the latter. In our case, EDX does not see boron and underestimates the amount of oxygen. We used in the EDX equipment an old-generation Si(Li) detector. For these reasons, the elemental composition was determined by the XPS.

In future works, we’ll try to add a mapping obtained by STEM.

(3) Line 232-233, “The Si 2s peak, located at 153.4 eV, corresponds to SiO2. The B 1s peak is chemically shifted to 192.4 eV, which means the existence of the B2O3 oxide”. Please mention the references for these measurements.

Response:

Many thanks for the helpful comment. We have added references and changed the chemical formula of silicon dioxide to the term "oxidized silicon". It is actually more of a silicate than a dioxide, as the reference BE values for silica range from 153.9 to 154.9 eV and for silicate from 153.0 to 154.2 eV. As to boron oxide, the obtained BE value of B 1s is in good agreement with the reference data.

(4) In the discussion of Figures 2, 3, and 4 authors have just illustrated the results, without describing the reasoning.

Response:

We presented a description and interpretation of the results illustrated in Figures 2-4 and showed that at the initial technological stage, layers with a high oxygen concentration were obtained, consisting of oxidized Fe, Si and B (there is no way to add Co since all cobalt lines are masked by overlapping with lines of other elements) and containing crystalline particles. The experimental conclusions that can be drawn from these results were discussed in the paragraph “Based on the data from XPS elemental analysis and TEM structural analysis…”, lines 255-262.

It is possible that this remark refers to the lack of argumentation of the XPS and TEM studies. So we have added a reasoned explanations at the beginning of subsection 3.2.

(5) Overall the research work is very well explained, please make the conclusion precise and remove the extra discussion sentences from conclusion.

Response

Thank you, we’ve improved the conclusions.

Reviewer 2 Report

Authors have successfully developed ME MEMS magnetic field sensor based on a laminated heterostructure of bidomain lithium niobate and metglas alloy.
ME materials for sensing application his an interesting topic with high impact, nevertheless some major issues must be solver before acceptance, namely:

a) English grammar and language should be revised;
b) Polymer-based ME materials should be included in the revision (namely the electronic optimization of metglas/pvdf for energy harvesting; the size effects on the ME response of pvdf/vitrovac composites; the greener production of ME PVDF-TrFE/CoFe2O4 composites and the printing of PVDF-based ME materials);
c) Why authors have measured the alfa32? Are they confusing the sample's geometry with the crystallographic directions?
d) Authors should include the piezoelectric characterization;
e) HACvsalfa coefficient should be included in the revised document. 

Author Response

Responses:

  1. As to the revision of the “English grammar and language”, the text has been proofread by an experienced professional before submission. Of course, we’ll carefully check the grammar and stylistics of the revised manuscript, too.
  2. The reviewer requires an extended citation of publications which only tangentially have to do with the special content of our paper.
  3. The ME coefficient has a 32 index to indicate that we measure the ME effect in a direction where the piezoelectric coefficient in the y+128º-cut LN is maximized, more detailed discussion can be found in DOI:10.1109/TUFFC.2017.2694342
  4. We measured the piezoelectric coefficient and deflection for all our samples before the metglas was deposited. The method is described in doi:10.1063/1.5038014 (suppl. mater.). Only the best bidomain crystal was used in the magnetoelectric measurements. However, the aim of the article is different and we did not present this part.
  5. Figure 9 presents VME vs. HAC which is almost the same as α32 vs. HAC. Such a graph would show a constant dependence up to the noise level, which does not make sense.

Reviewer 3 Report

This study proposes a magnetoelectric MEMS magnetic field sensor based on a laminated heterostructure of bidomain lithium niobate and Metglas. I have the following comments:

(1) The authors propose a technique creating ferroelectric bidomain structure, it is suggested a concise summary or flow chart consisting of sequential steps are included in the section of Materials and Methods.

(2) How is the performance of magnetic sensor compared to the similar buck laminates consisting of lithium niobate? Please add the comments and explanations.

(3) For Table 2, the Q factor of propose cantilever typed ME device seems lower than other reports, why the limit of detection is even better? More detailed analyses are needed.

Author Response

Reviewer: This study proposes a magnetoelectric MEMS magnetic field sensor based on a laminated heterostructure of bidomain lithium niobate and Metglas. I have the following comments:

(1) The authors propose a technique creating ferroelectric bidomain structure, it is suggested a concise summary or flow chart consisting of sequential steps are included in the section of Materials and Methods.

Response:

The authors thank the Reviewer for the comment. The process that we used for the formation of the bidomain ferroelectric structure in lithium niobate is based on the out-diffusion of Li2O. A gradient of point defects in a crystal plate leads to domain switching. The method is well known and described in the literature (we added reference [27] to the review of different approaches to the production of bidomain crystals and their possible applications).

We also added more details about the annealing of the LN crystals to the section of Materials and Methods.

“The structures were cleaned after the microblasting process and annealed for 5 hours at a temperature of 1100ºC in an air atmosphere, thus, conditions were created for the formation of a bidomain ferroelectric structure of the head-to-head type. The method we used is based on the evaporation of lithium oxide out of the surface of the LN crystal [27] (so-called out-diffusion annealing). The crystals were placed in a muffle furnace on a sapphire wafer using narrow and thin (0.5 mm) sapphire spacers, thus a symmetrical out-diffusion of Li2O occurred.”

(2) How is the performance of magnetic sensor compared to the similar buck laminates consisting of lithium niobate? Please add the comments and explanations.

Response:

Thank you for the helpful comment. In the introduction we added our best result of sensitivity to magnetic field measured by an ME composite based on bulk bidomain LN (ref. [36], line 109), however, the results are difficult to compare, because the record value of sensitivity was measured at a higher frequency (6.7 kHz), also the ratio of the thicknesses of the magnetostrictive to piezoelectric layer was higher (0.06) than in the presented work (0.025). Furthermore, the sample in the article [36] was fixed in the center when in the present article we investigated the cantilever. Most previous works were made on composite samples which were prepared using epoxy glue to connect metglas and bulk LN crystal.

(3) For Table 2, the Q factor of propose cantilever typed ME device seems lower than other reports, why the limit of detection is even better? More detailed analyses are needed.

Response:

Thank you for your question.

The sensitivity of ME sensors is defined as the voltage noise density (or noise level) and conversion coefficient equal to α·t, where t is the thickness of the PE phase:

S = Vn/(α·tPE) = Vn/(conversion coefficient) [T/Hz1/2].

α is proportional to the Q-factor.

Therefore, the sensitivity depends on many factors such as the noise level, Q-factor, conversion coefficient, etc. In the manuscript, we discussed (lines 397-400) that in our opinion the main limitation to the magnetic field sensitivity is the environmental vibrations readily exciting the bending resonance together with the AC magnetic field.

In our composite ME sample, we measured a higher conversion coefficient and a similar Q-factor in comparison to other works (Table 2). This combined with the low noise level made it possible to achieve a higher sensitivity.

Round 2

Reviewer 2 Report

The authors did not follow the suggestions.